# Detecting Mild Cognitive Impairment Through Digitized Trail-Making Test Interface

Raniero Lara-Garduno*  
Texas A&M University

Yajun Jia†  
Texas A&M University

Nicolaas E. Deutz‡  
Texas A&M University

Marielle Engelen§  
Texas A&M University

Nancy Leslie¶

Tracy Hammond‖  
Texas A&M University

## ABSTRACT

With the number of Alzheimer's patients reaching 5 million in 2014 according to the U.S. Center for Disease Control and Prevention, increasing emphasis has been placed on identifying and understanding its precursor condition, Mild Cognitive Impairment (MCI). MCI is characterized by subtle but abnormal cognitive decline and is challenging to detect without formal testing. Neuropsychologists use paper-and-pencil tests such as the Trail-Making Test (TMT) for diagnosis, and ongoing research places importance on high-granularity sketch data from digital TMTs. We present SmartStrokes, a digital TMT app designed to simulate the paper-and-pencil testing experience on a tablet and stylus. Our contribution frames the principles of digital sketch recognition and Human-Computer Interaction (HCI) into the existing neuropsychological test, outlining the creation of a pair of classification models that identify MCI on an individual segmented line basis. Such a per-line classification method which could provide localized sketching behavior indicative of MCI. We also present an interface for the digital TMT and a refinement of line segmentation algorithms from previous research to better distinguish between the actions that a participant takes when completing the exam.

**Index Terms:** Applied computing—Health Informatics; Human-centered computing—Human computer interaction; Human-centered computing—Tablet computers

## 1 INTRODUCTION

The U.S. Center for Disease Control and Prevention has reported 5 million Alzheimer's patients in 2014, with the expected number to more than double to 13.9 million by 2060. Due to advancements in interventions aimed at mild-to-moderate cases of Alzheimer's disease, neuropsychologists have placed an increasing emphasis on early detection of Mild Cognitive Impairment (MCI) to better preserve quality of life [4, 20, 34]. A clinical neuropsychologist typically conducts paper-and-pencil cognitive examinations on a patient to help detect MCI. This process is historically laborious, requires multiple rounds of testing, and frequently requires non-standardized subjective analysis of a patient's subtle behavioral patterns. Digitizing these clinical examinations, specifically the Trail-Making Test among them, has allowed researchers to attempt to aid the diagnosis process by employing machine learning for behavioral analysis. Existing work in this space has not yet fully leveraged recognition

---

*e-mail: raniero@LGinbox.com  
†e-mail: jia560@tamu.edu  
‡e-mail: nep.deutz@tamu.edu  
§e-mail: mpkj.engelen@ctral.org  
¶e-mail: nleslie.phd@gmail.com  
‖e-mail: hammond@tamu.edu

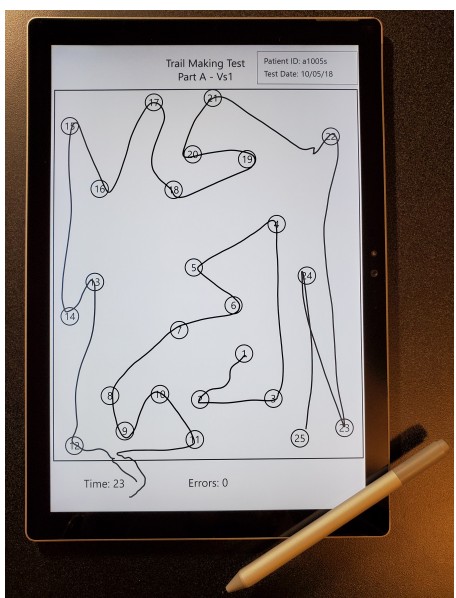

Figure 1: A sample completed test in our SmartStrokes app. The interface is designed to be as close as possible to an actual paper-and-pencil test

techniques used in digital sketch recognition, particularly research that links sketching with cognition. In particular, the application of HCI principles to detect MCI via digitized testing interfaces in the context of neuropsychology is a topic we believe has not yet fully explored. Our contribution presented in this paper is to integrate HCI and digital sketch recognition into the domain of neuropsychology to deliver more granular recognition on a digitized TMT. We analyze and classify individual test segments rather than the more traditional method of one determination for an entire test. We also discuss the limitations and potential avenues for future research that surfaced during the completion of this research.

### 1.1 Mild Cognitive Impairment

The characteristics of MCI were initially established as part of the Global Deterioration Scale (DGS) [47], defining it as a syndrome where an individual's cognitive decline is greater than expected for their age [24, 48]. It is considered to be a precursor to more severe cognitive decline that may advance into dementia, with Alzheimer's in particular being likely. The existence of MCI in itself, however, is not indicative that cognitive will necessarily decline further, as the cognition of many MCI patients never develops into dementia. Additionally, unlike these more severe forms of cognitive decline, MCI does not severely impact one's daily quality of life [64] and can thus be challenging to diagnose. This means that often the signs are

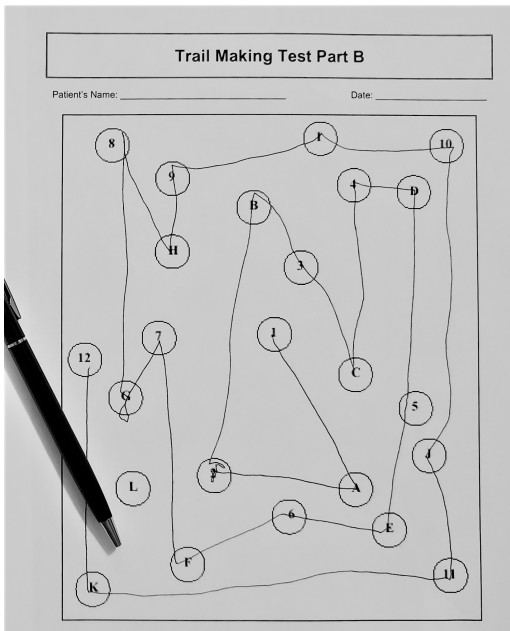

Figure 2: Sample of traditional paper-and-pencil versions of Trail-Making Test B

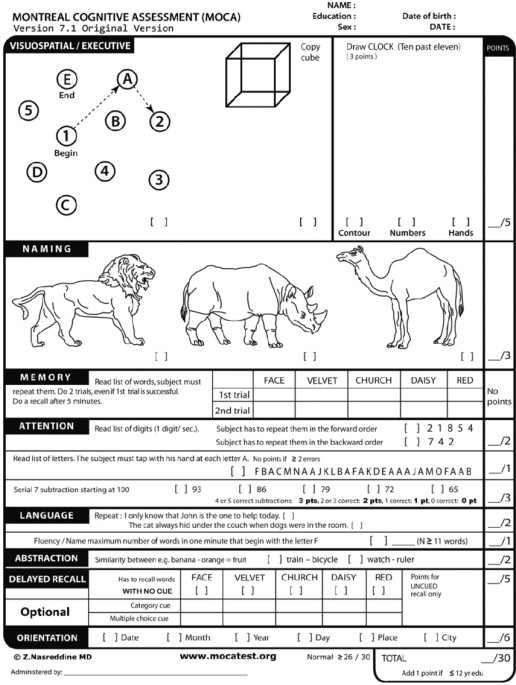

Figure 3: The Montreal Cognitive Assessment (MoCA). Image courtesy of mocatest.org [36]

subtle and can be easily dismissed as expected decline in executive function for an individual's age. MCI itself is characterized as not having a significant impact on daily activities, and may not manifest in a noticeable way for years, making it difficult to definitively diagnose and track.

In cases where MCI does worsen, the characteristics of severe cognitive decline can vary depending on background and genetic conditions particular to the patient. Reisberg *et al.* [47] specifies the emergence of "behavioral disturbances", neurological abnormalities, electrophysiological changes, motor deficits, balance and coordination deficits, and general active daily living activity deficits. With an increase in life expectancy correlating to a rise in the prevalence of dementia and Alzheimer's disease, research attention has turned to the successful identification of MCI and how existing tools can be improved to assist.

## 1.2 Trail-Making Test

Clinicians have historically relied on paper-and-pencil neuropsychological examinations as one of the primary methods to diagnose MCI. These typically involve a series of simple tasks for an individual to complete, and have been shown to be sensitive to the same cognitive functions affected by MCI through several decades of research [6]. We focus on Trail-Making Test, a connect-the-dots task that tests executive function and active memory. Initially conceived as a test to assess general intelligence, the TMT is known to be sensitive to cognitive decline and possible early signs of dementia. Currently the Trail-Making Test is widely used in neuropsychologists' test battery to assess for various signs of cognitive decline, including MCI [59]. The switching between numbers and letters found in the TMT-B relies on frontal lobe function [12, 26, 29, 40, 53], and is one of the primary reasons for the sensitivity of the test to MCI.

The test consists of two separate connect-the-dots tasks. The participant is handed a piece of paper with a series of labeled dots printed on it, and are handed a pen or pencil with which to connect the dots. The A variant of this test consists of a participant connecting dots in ascending numerical order (1, 2, 3, and so on), while the B variant of the test requires connecting dots alternating between numbers and letters in ascending order (1, A, 2, B, and so on). The

participant is typically asked to complete the test as a pair, starting with variant A and immediately followed by variant B. Multiple layouts of these tests exist and are used when a clinician wishes to test the participant more than once, since a different arrangement of labeled dots is necessary to avoid the learning effect. Dot layout has been observed to directly affect time to completion on healthy populations [5]. Participants are asked to not lift their pen or pencil whenever possible, or when they connect to the wrong dot and must return to the previous dot.

Assessment of Trail-Making Tests is primarily done in two ways: comparing the test score with established normative data, and the qualitatively observed behavior of a participant as the test is being completed. The test score is calculated as the test's time to completion rounded to the nearest whole number. The fact that a score is reported as a single numerical value necessitated the qualitative observation, and over the decades clinicians have devised multiple methods for assessing a participant's performance as they complete the test. Colored pencils, video recordings, and observing behaviors from sitting posture to the way the patient holds their pen are just some of the qualitative observations made by clinicians.

These measures taken highlight the notion that the behavior a participant exhibits during the test is just as important, if not more so, as the single time-to-completion reported score. The subtle nature of MCI, however, has historically meant that clinicians rely on their own expertise and experience for qualitative observations. Most recent advancements to digital sketching technology have made it feasible for these tests to be assessed with much higher granularity than in previous decades, but research aiming to capitalize on this feasibility is limited.

## 1.3 The Montreal Cognitive Assessment

The Montreal Cognitive Assessment (MoCA) is among the most widely used assessment protocols for gauging an individual's cognitive function. It consists of various short tasks, both written and verbal, aimed at testing various functions of a person's cognition and is frequently administered as a triage to help determine whether

a patient requires further diagnosis and possible treatment, and is also frequently used to determine whether patients have symptoms of MCI [39]. Originally developed in 1995 by Ziad Nasreddine [23], it has since been the subject of various validation studies [26, 57]. Normative data for the MoCA has been collected and analyzed for patients of various populations [38, 54], diseases [11, 45], cognitive states [11, 25], and post-trauma conditions [21, 60]. The primary conditions that it has been validated for include MCI, Alzheimer's disease and Parkinson's Disease dementias [14, 39, 57], and has been shown to be more sensitive to MCI-related decline than other examinations such as the Mini-Mental State Examination (MMSE) [49]. Hobson describes that the MoCA can assess cognitive domains including but not limited to "Visuospatial/Executive, Naming, Memory, Attention, Language, Abstraction, Delayed Recall and Orientation (to time and place)" [23].

The MoCA is frequently used in tandem with other neuropsychological examinations for ensuring that its results are consistent across various other examinations such as the Trail-Making Test. In effect, one might consider performance of the MoCA and the TMT to be correlated, such that exceptionally well or poor performance of one test is likely to lead to poor performance in the other. Indeed, a brief TMT-B appears in the MoCa as one of the tasks [26], and they both test of the same frontal lobe function.

## 2 RELATED WORK

### 2.1 Cognition in Digital Sketch Recognition

One of the prevalent methods of digital sketch recognition is through the analysis of sketches as "gestures" comprising of geometric properties of sketches. This includes but is not limited to line length, speed, acceleration, line straightness, and various trigonometric properties of line strokes. Individual features were calculated in early efforts from Rubine *et al.* [50, 51], and later expanded by Stahovich *et al.* [10, 27, 55], Long *et al.* [31], Paulson *et al.* [43, 44], and Alamudum *et al.* [3]. Digitial sketch recognition initially leveraged machine learning to afford developers tools to recognize simple geometric shapes. Shape recognition expanded to alphabets, scaffolded recognition to identify components of complex composite shapes, and entire sketches. Machine learning algorithms have allowed these analyses to be made feasible over a large corpus, resulting in models that are able to distinguish between objects depending on subtle changes in sketching behavior.

An increasingly common application of digital sketch recognition does not identify the shapes drawn, but rather characteristics of those who draw them. Kim *et al.* identified strong correlations between sketching behavior and early cognitive development in infants [28]. Davis *et al.* [63] and Muller *et al.* [37] similarly has focused on cognitive decline by analyzing sketches from Clock-Drawing tests [58]. Zham *et al.* identified the presence of Parkison's disease through the way a participant drew spirals with a smart-pen [63]. Digital variants on existing neuropsychological tests are numerous, with various proposed systems designed for test automation, diagnosis assistance, or self-administration [7, 18, 52, 62]

### 2.2 Digitized Trail-Making Tests

Multiple computerized variations on the Trail-Making Test have been developed and studied [22]. Drapeau *et al.* noted the clear difference in performance between a paper-and-pencil TMT and a digitized version completed with a computer mouse [16]. Jager *et al.* directly studied differences in performances between paper-and-pencil and computerized neuropsychological tests [15]. Smith *et al.* explored the possibility of implementing several cognitive testing tools with mobile technology [56]. Prange *et al.* uses a large amount of digital sketch recognition features to classify participants as "healthy" or "suspicious" [46] but does not heavily anchor the features on neuropsychological and HCI principles nor is there a granular per-line analysis made beyond determining whether a

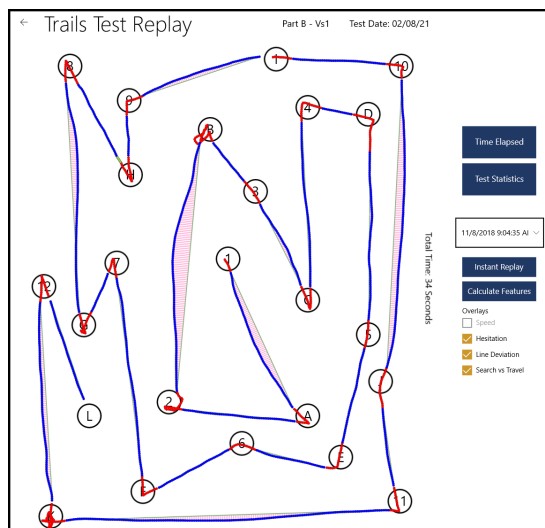

Figure 4: Test analysis interface of *SmartStrokes*, demonstrating line deviation and separation of search and travel lines on a completed test

line connects two dots. More specifically in the tele-health space, Brehmer *et al.* contextualized the challenges and considerations to be taken with implementing computerized neuropsychological exams at home where there could be interruptions [9]. More novel research in this includes the work of Lara-Garduno et al. who presents a touch-based novel neuropsychological examination based on the TMT [30]. With the advancement and increasing affordability of pen and touch technology and mobile computing, interest turned to digitizing these tests to simulate the original pen-and-paper experience.

One of the most recent attempts at digitizing the Trail-Making Test and leveraging machine learning to aid in the diagnosis process comes from the work of Dahmen *et al.* [13]. This work involved the use of a tablet and stylus to re-create the Trail-Making Test and a user study consisting of (N=54) older adult participants. Digital sketches were used as traning data to predict two types of assessment: that same participant's scores of the Telephone Interview for Cognitive Status (TICS) [8] and Frontal Assessment Battery (FAB) [17] performance scores, and the prediction of a participant's condition as "healthy" or "neurologic". Prediction of a participant's condition using features mostly focusing on dwell-time yielded accuracies ranging from 44% and 67%. Predictions were done on a per-test basis using feature averages rather than localizing or segmenting lines.

### 2.3 Proposed Contribution

Our proposed contribution presents an interface that collects digital sketch that that is then used to create a classification model to distinguish between MCI and healthy participants on a **per-line** basis. It builds on existing work from Dahmen *et al.* [13], which in its conclusion states the belief that the high-granularity digital sketch data from a digitized version of the TMT could provide higher-granularity analysis. Per-line classification would offer two advantages: 1) targeting individual lines for classification of MCI could give a more localized assessment of individual dots that challenged the participant, and 2) building a classification model that analyzes sketches on a per-line basis could allow generalizability for more layouts, since as previously mentioned the TMT frequently needs a wide variety of dot layouts to avoid the learning effect.

Further, our proposed contribution uses the scores from the Montreal Cognitive Assessment (MoCA) to detect MCI, whereas the existing work from Dahmen uses the TICS and FAB to assess more

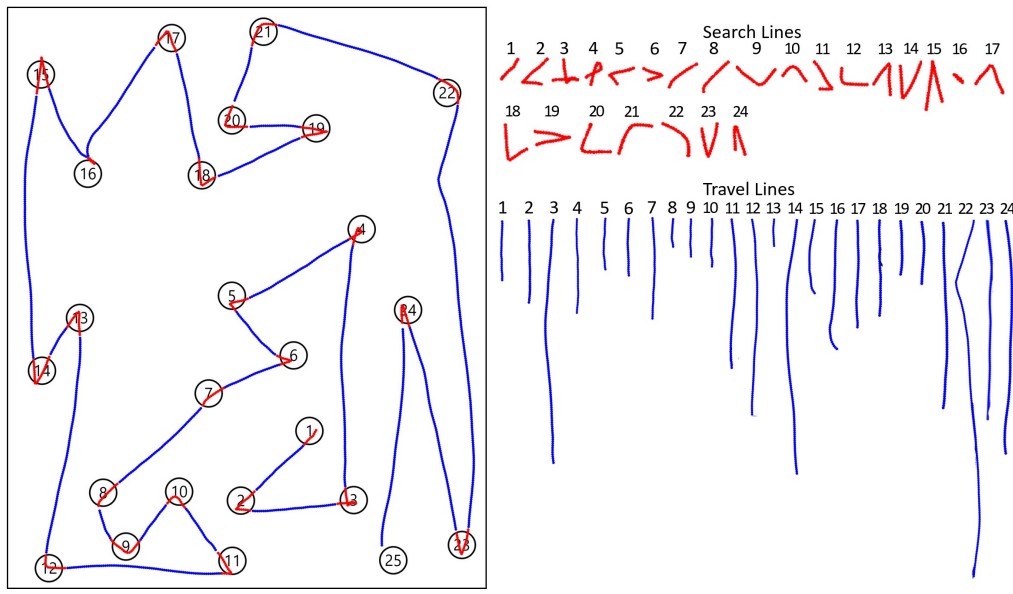

Figure 5: Separated travel and search lines. Travel lines are rotated to always face a top-to-bottom orientation.

advanced dementia. MCI is characterized as being much more subtle in nature, meaning milder cases of MCI frequently result in sketching behavior that only slightly deviates from a healthy participant. Our proposed solution achieves an accuracy similar to Dahmen's existing work, with the added advantages of offering classifications on a more granular per-line basis, and classifying for a more subtle degree of cognitive decline.

## 3 INTERFACE DESIGN

*SmartStrokes* is a digital testing suite focused on re-creating commonly-used Trail-Making Test layouts for use on Microsoft Surface Pro 4 devices. The Universal Windows Platform (UWP) was chosen for reasons that include rapid development of a mobile-style application on Windows devices, ease of exporting pen data for analysis, and its firmware-level digital pen integration with Surface Pen devices and allows us to easily extract pen pressure to supplement our feature set.

The the system has two simultaneous end users: medical and other personnel who proctor the exam (referred to in this paper as "proctors"), and participants who complete the test (referred to as "participants"). Every proctor is associated with individual proctor IDs and every test is directly associated with each participant. All participant data including completed tests can only be accessed from within the app if the test proctor username and password is entered at the login screen. Proctors have the option to export digital images of the completed examinations at the conclusion of each test, at which point the proctor should ensure proper data anonymization practices such as ensuring the file names and location do not contain identifiable information.

A total of 8 separate Trail-Making Test layouts were converted into a digital format, comprising of 4 pairs of the A and B variations. These Trail-Making Test layouts are among those that are generally used by neuropsychologists when conducting these tests in their practice. Dimensions of the white space were cropped to account for the different aspect ratio between the Surface Pro 4 and a regular 8" 1/2 x 11" piece of paper, and the layout and size of the dots were scaled accordingly. The test interface itself resembles a paper-and-pen test as much as possible. This includes extending the drawing canvas across the entire screen, beyond the black large rectangle where the dots are placed; on a real piece of paper some partici-

pants may draw outside of the large rectangle despite it not being advised to do so. Our intention with this interface is to capture the same types of mistakes a participant might make with a traditional pencil-and-paper modality. SmartStrokes also intentionally offers no indication of visual feedback given to participants when the next dot is connected in sequence. An earlier version of the test turned correctly connected dots green, but experts advisors suggested that feedback be only given in the case of a mistake since that is the only scenario in which a clinician would intervene. Although testing protocol dictates participants should only complete each pair once to avoid the Learning Effect, *SmartStrokes* has the ability to test each participant as many times as they wish on any arbitrary layout and order to accommodate for any testing procedure.

Completed tests can be viewed at any time if the application is signed into the proctor's profile. The time-series sketching data allows proctors to review each participant's tests at their leisure and can also choose to replay the test in real-time to qualitatively review the participant's performance. Additionally, *SmartStrokes* can display color-coded visualizations of the sketch that include: separation of **travel** and '**Search** actions during the test, pen speed, pressure, location of "hesitation" regions, and line straightness.

*SmartStrokes* also assists in data analysis by performing feature calculation of individual tests and outputting the anonymized data into a local Comma-Separated Value file (CSV). Additionally, the proctor can choose to automatically perform this calculation for every test associated with that proctor. This allows proctors to conduct data analytics by easily importing the CSV for rapid visualization and machine-learning analytics tasks.

## 4 ANALYZING DIGITIZED TRAIL-MAKING TESTS

One of the significant challenges in analyzing the Trail-Making Test is in the proper segmentation of the data. Although the task is designed to result in simple straight lines, the ideal resulting sketch consists of a singular line making 25 stops that change direction each time. Analysis is further complicated by behaviors arising from cognitive decline, most commonly involving repeated mistakes and prolonged periods of searching for the next dot, hesitation, or doubt.

Complicated line drawings are frequently segmented in the digital sketch recognition domain in order to properly characterize key elements in the sketch. The most appropriate domain-specific method

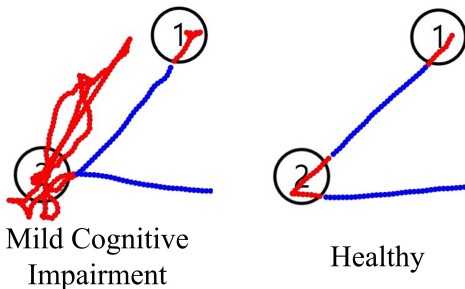

Figure 6: A clear example of the search Line difference between an MCI participant and a healthy one. This discrepancy is usually the result of the participant unable to locate the next dot in the sequence for an extended period of time. Although the discrepancy is obvious in this example, not all MCI participants exhibit this behavior, making diagnosis challenging.

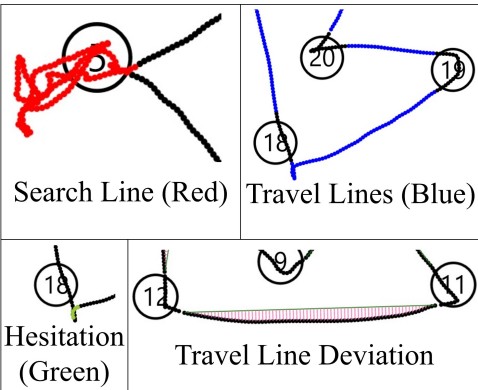

Figure 7: Four of the color-coded features and sketch properties that *SmartStrokes* can display. Search and travel Lines are also used to segment data for constructing the classificaton models

of line segmentation separates the lines in two different categories: **Search** lines, and **Travel** lines; **Search** lines are all lines drawn when the participant is looking for the next dot, and **Travel** lines are the line segments where the participant is actively moving from one dot to the next

The following two subsections outline the differences between the two types of lines, what thresholds exist between the line segmentation, and which sketching characteristics we believed would be the most relevant to identifying MCI.

### 4.1 Search Lines

According to the protocol of Trail-Making tests as outlined by the Compendium of Neuropsychological Examinations [59], participants are required to have their pen on the test at all times even when not moving between dots. This is done for two reasons. The first is that it is less likely that a participant loses their place if they do not lift their pen as they search for their next dot. The second reason is that this maximizes the data collected, since a participant who leaves their pen on the paper as they search for the next dot almost always results in randomized pen movements while they move their hand to see the rest of the test. This kind of sketching is typically characterized by noisy, erratic movement that tends to meander around the current dot as the participant searches for the next one. This is the kind of line that we identify as a **search** line.

We define the beginning of a search line as the instant a participant enters the next correct dot in the TMT sequence. We define the end of the search line as the moment the participant identifies the next dot and moves out of the area of the current dot. We complicate the definition of the end of the search line beyond simply "outside of a dot", because of how participants behave when searching for a dot for a long time; participants who meander around a dot for a long time frequently move the pen inside and outside of the dot's area as they look for the next dot in the sequence. They may also stray away from the dot before identifying the next one. For these reasons we include an additional speed threshold outside of a dot's area as the end of the search line segment.

Healthy participants typically do not pause for long as they search for the next dot in the sequence, with some participants not pausing at all. Indeed, search lines from typical healthy participants are usually shorter in length and have a single curve clearly detailing the change in direction from the previous dot in the sequence to the next with very little or no meandering behavior. MCI participants or any other participants who find the TMT challenging typically remain in this search state for longer, resulting in longer and more erratic search line segments. Figure 6 shows such an example where

an MCI participants' search state results in a significantly longer and more meandering search line.

### 4.2 Travel Lines

Our parameters for defining travel lines are more straightforward, as we define travel lines as the moment the participant begins to move with intent to arrive at the next dot, and the travel line segment ends when the next dot in the sequence is reached. When done correctly, the travel line will be a single straight line from the previous dot to the next. We implemented a pen speed threshold to identify this "intent to move" to help us clearly delineate between a search line outside of a dot's area and the moment the participant moves toward the next.

Every dot in a Trail-Making Test in sequence can be connected with a single straight line. For that reason, participants who perform well in the TMT usually have a series of travel lines drawn straighter and without turning to change direction while moving from one dot to the next. Participants who perform poorly sometimes stop in the middle of a **Travel** line to either check their destination again or change direction as they realize they are going to the wrong dot.

Sometimes, such participants stop entirely in the middle of travel and begin a similar **Search** behavior to find the next dot. We call all of these mid-travel stops or significant reduction in speed as "hesitation". While not every participant with MCI enters this state, several instances of these hesitation states in one test is likely to point to a poorly-performing examination.

### 5 DATA COLLECTION AND ANALYSIS

This subsection details the process by which TMT data collection was conducted, and the sketch recognition features that were selected and applied to a machine-learning classification model to detect MCI.

### 5.1 Data Collection

37 participants were recruited for data collection and classification purposes. Participants were screened and classified as MCI or healthy based on scores from the Montreal Cognitive Assessment (MoCA) [39], with the MoCA scores ranging from 0 to 30. The inclusion criteria for participants were the following:

- **Healthy subjects without MCI**: Healthy older adults, normal cognition. MoCA score is 26 or above. Subject group labeled as "Healthy" for model classification purposes.

- **Healthy subjects with MCI**: Healthy older adults. MoCA score is between 19 and 26. Subject group labeled as "MCI" for model classification purposes.

Table 1: Participant demographics for user study. 95% confidence interval for participant age is $71.43 \pm 2.41$, for MoCA scores $24.54 \pm 0.91$

| Age Range | Male | Female | MCI | Non-MCI | Avg. Age | Avg. MoCA |
|---|---|---|---|---|---|---|
| 55-59 | 1 | 1 | 0 | 2 | 57 | 26.5 |
| 60-64 | 1 | 5 | 1 | 5 | 63 | 26.5 |
| 65-69 | 5 | 2 | 5 | 2 | 67.7 | 23.9 |
| 70-74 | 3 | 5 | 4 | 4 | 71.5 | 25.5 |
| 75-79 | 4 | 5 | 7 | 2 | 76 | 23.3 |
| 80-84 | 2 | 0 | 1 | 1 | 82 | 25.5 |
| 85-89 | 2 | 1 | 3 | 0 | 85.7 | 21.3 |
| Totals | 18 | 19 | 21 | 16 | 71.43 | 24.54 |

- **Exclusion criteria**: Older adult subjects with a MoCA score below 19; any history of severe medical/neurological/psychiatric disease, including diabetes/hypertension; taking medication primarily targeting central nervous system; any other condition at investigator's judgment that clearly demonstrates severe cognitive decline

Additional demographic information is available on Table 1.

All participants were recruited from a known pool of potential candidates, doctor referrals to this study, as well as open calls for participants via email. At the time of recruitment a pre-screening was conducted to ensure that the participants were not situated outside of the inclusion criteria. We administered a MoCA test to each potential candidate and is graded afterwards. If the candidate satisfied the inclusion criteria for one of the two possible categories of "with MCI" or "without MCI", a secondary visit was scheduled was scheduled to start at 8 AM and participants were asked to return as well-rested as possible.

At the time of the data collection procedure, all participants were given two sets of Trail-Making. Each test set consisted of a Trails A variant (numbers) and its accompanying Trails B variant (alternating numbers and letters). Each of the two sets used different standard dot layouts to eliminate a learning effect [35]. All participants were given the same Microsoft Surface Pro 3 device with accompanying Surface Pen to complete the digital tests. Participants were asked to connect the dots in ascending order as per the instructions detailed in the Compendium for Neuropsychological Examinations [59].

As *SmartStrokes* provides minimal feedback of mistakes so as to simulate the paper-and-pencil test taking experience, the test proctors similarly followed paper-and-pencil procedures that include notifying the participant whenever a mistake was made, but the participant is otherwise left to analyze the layout and make corrections to mistakes. Participants were instructed to place their pen down on the last correct labeled dot and try again. While we save the lines that were drawn to connect incorrect dots, those lines are made invisible in real-time while taking the test.

For the purposes of classification we refer to our subjects without MCI as "healthy", meaning subjects in the first category of participants. However, we must highlight that our MCI participants are not considered "unhealthy" by contrast. Indeed, MCI is considered a precursor to severe conditions such as Alzheimer's and dementia, and people with this condition are still considered "healthy" by every metric (see Section 1.1). In order to study the effects of a possible change of sketching behaviors, however, we elected to consider the two possible conditions as "healthy, without MCI", and "healthy, with MCI".

## 5.2 Preprocessing

Several pre-processing steps are conducted on individual completed examinations. Each test's sketch data is separated by travel and search lines according to the description in Section 4. Sketch data is then resampled to uniform interspace $S$ using the formula:

$$S = \frac{\sqrt{(x_m - x_n)^2 + (y_m - y_n)^2}}{c} \quad (1)$$

Where $S$ is the new spacing between each sample, $(x_m, y_m)$ is the lower-right corner of the sketch $(x_n, y_n)$ is the upper-left corner of the sketch, and c is an empirically derived constant $c = 40$ that is frequently used in the domain of digital sketch recognition for optimal distance between samples that balances high enough granularity for feature calculation with few enough samples for computational efficiency.

Lastly, we implemented an additional key step in this process by normalizing individual line rotation for travel lines. The chosen features explained later in this section make significant use of sketch direction, either as a per-sample basis or the entire line. In more typical digital sketch recognition problems, features relating to direction inform a participant's style of drawing, or are directly related to the type of shape that the participant intends to draw. The Trail-Making Test, however, places all dots in pre-arranged locations that strongly influences the direction of a correct line. This would introduce a confounder, since differences between angles or sketch direction would not be attributed to MCI but rather the layout of the test's dots. We normalize travel lines by rotating every line such that the endpoint of the line is directly underneath the start point. This allows us to still be able to leverage direction-related sketch features to calculate characteristics like tremor, changes in direction due to mistakes made, and other types of directionality affected by the participant's performance rather than the layout of the Trail-Making Test. We are not aware of similar work in constructing a Trail-Making Test classification model that employs this segmented line direction normalization technique. To account for a physical range of motion confounder, participants were observed to ensure that they did not have physical difficulty in moving in a particular direction. To that end we observed no difficulties in participants nor did any participant report one themselves.

## 5.3 Feature Calculation

### 5.3.1 Rubine Features

We implemented a combination of digital sketch recognition features known to yield accurate models in similar research projects. The first set of 13 features introduced by Rubine *et al.*, abbreviated as "Rubine features" [50]. The 13 features were first introduced alongside a recognition technology named GRANDMA (Gesture Recognizers Automated in a Novel Direct Manipulation Architecture), a toolkit that sought to provide end users with the ability to train any gesture for recognition using a click-and-drag interface. The Rubine features themselves have since then been implemented in various sketch recognition projects that can gauge not only the type of shape that is drawn, but also the cognitive state of the participant who drew them. Rubine features $f_1$ and $f_2$ specify the cosine and sine features of the first few samples, usually limited to the first two samples as was done in our implementation. The bounding box diagonal of the entire gesture is analyzed as features $f_3$ and $f_4$. The distance in pixels between the first and the last point is specified in feature $f_5$. The difference between the first and last point of a gesture is analyzed through features $f_6$ cosine and $f_7$ sine between the start and end points, the total length of the gesture is calculated for $f_8$, and the total angle traversed is $f_9$. Three total summations are calculated, with $f_9$ being the total angle traversed over the course of the gesture, $f_{10}$ being the sum of the absolute value of the angle per mouse point that does not take into account direction, and $f_{11}$ being the sum of the square of the value of $f_9$. The square of the maximum speed achieved in the gesture is $f_{12}$, and the last feature $f_{13}$ is the total duration of the gesture, measured in milliseconds. The calculations for the Rubine features are provided on Table 2.

Table 2: Rubine features $f_1$ through $f_{13}$. Let $\Delta x_p = x_{p+1} - x_p$, and $\Delta y_p = y_{p+1} - y_p$, and $\Delta t_p = t_{p+1} - t_p$

| Rubine Features | |
|---|---|
| $f_1 = \dfrac{x_2 - x_0}{\sqrt{(x_2-x_0)^2 + (y_2-y_0)^2}}$ | $f_8 = \sum_{p=1}^{P-2} \sqrt{\Delta x_p^2 + \Delta y_p^2}$ |
| $f_2 = \dfrac{y_2 - y_0}{\sqrt{(x_2-x_0)^2 + (y_2-y_0)^2}}$ | $f_9 = \sum_{p=1}^{P-2} \theta_p$ |
| $f_3 = \sqrt{(x_{mx} - x_{mn})^2 + (y_{mx} - y_{mn})^2}$ | $f_{10} = \sum_{p=1}^{P-2} |\theta_p|$ |
| $f_4 = arctan\dfrac{y_{max} - y_{min}}{x_{max} - x_{min}}$ | $f_{11} = \sum_{p=1}^{P-2} \theta_p^2$ |
| $f_5 = \sqrt{(x_{p-1} - x_0)^2 + (y_{p-1} - y_0)^2}$ | $f_{12} = \max_{p=0}^{P-2} \dfrac{\Delta x_p^2 + \Delta y_p^2}{\Delta t_p^2}$ |
| $f_6 = \dfrac{(x_{p-1} - x_0)}{f_5}$ | $f_{13} = t_{P-1} - t_0$ |
| $f_7 = \dfrac{(y_{p-1} - y_0)}{f_5}$ | |
| $\theta_p = arctan\dfrac{\Delta x_p \Delta y_{p-1} - \Delta x_{p-1} \Delta y_p}{\Delta x_p \Delta x_{p-1} + \Delta y_p \Delta y_{p-1}}$ | |

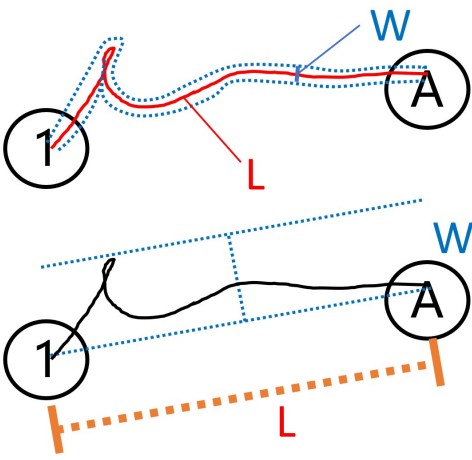

Figure 8: The traditional application of the Steering Law is on top, with $W$ and $L$ being predetermined. Our use of Steering Law, on bottom, creates a simple tunnel with $W$ based on the total "width" of the pen trajectory.

The Rubine features represent the various geometric properties of any given gesture. They can measure speed, curvature, direction at the start and ends of the gesture, total time taken, and the properties of the total area (referred to by Rubine as the "bounding box") of any particular gesture. These features offer an alternative to template-matching recognition in that they do not require a point-for-point comparison, but rather are geometric calculations of the gestures themselves. Although these have been used mostly for recognizing gestures, their frequent use in recognizing shapes provides us with an opportunity for analysis of cognitive impairment.

### 5.3.2 Fitts' and Steering Law Features

We leverage principles from Fitts' Law by calculating that law's Index of Difficulty [32]:

$$ID_F = \log_2 \frac{2D}{W} \tag{2}$$

Fitts' Law was originally conceived as a method to quantify complexity [19] and has has been widely used in HCI research, particularly UI navigation tasks [33]. Fitts' Law is rooted in tracing lines across distances between targets and measures that task's complexity into measures of performance, which we believe could be leveraged to help identify task performance.

A related feature we use is the more recent variant, the Steering Law. The Steering Law assesses the difficulty of a participant navigating a pointer through a path with a set width [1, 2]. For a generic tunnel $C$, and a width $W(s)$ along the path, the Steering Law's Index of Difficulty $ID_S$ is:

$$ID_S = \int_C \frac{ds}{W(s)} \tag{3}$$

For our purposes, we use a straight path of length $L$ and a constant with $W$ as defined by Pastel *et al.* [41], which reduces $ID_S$ to:

$$ID_S = \frac{L}{W} \tag{4}$$

By using the participant's input lines as the basis for calculating $W$, we essentially create a form of performance index using the Steering Law. For the Trail-Making Test, a narrower line width $W$ is straighter and effectively more difficult to recreate. We integrated this metric as a feature for the classification model to test whether a participant with MCI would create lines with a generally lower $ID_S$. We also scaled and averaged $ID_F$ and $ID_S$ as a separate feature to explore a possible combination of the two. It is reported in Table 3 as *fittsSteering*.

### 5.3.3 Additional Behavioral Features

*Hesitation* is a feature that we briefly discussed in section – as a feature unique to travel lines. It characterizes the prevalence of stop-and-go motion for participants who start connecting a dot but stop or slow down significantly while inside a travel state. *Hesitation* begins when the pen slows down to an empirically-derived speed of 0.4 over five consecutive sampled points, and our calculated feature is distance the pen traveled while the pen remains in this state. The pen exits this state when at least five consecutive sampled points have a speed above 0.4. This threshold was determined when observing participants during pilot studies, where we sought to capture the most accurate subset of drawn lines during the time that participants hesitated when observing the need to change direction. The threshold was refined over a series of iterations to most accurately capture the hesitation state. If the pen enters *Hesitation* state multiple times inside a single travel line, the total distance across all of these states is reported for the one travel line

*Line Ratio* is a feature meant to normalize the length of a participant's drawn line. We believe the length of the line is important to understand how confident and accurate the lines were connected, since meandering behaviors and course correction would naturally result in a longer line than a straight line drawn directly from dot to dot. However, a drawn line will also be longer if the correct dots are placed further apart. The Trail-Making Test is explicitly designed to place dots a variety of distances from each other to measure a participant's ability to identify dots that might be further away from their immediate location. To take *relative* line length into account we divide the total distance drawn from one dot to the next by the theoretical "perfect" line drawn from one dot to the next. The closer the number is to 1, the closer to "perfect" this distance becomes and the better a participant performs. The formula for Line Ratio $R_{ln}$ is found below, where $(x_n, y_n)$ is the final sampled dot of the input line:

$$R_{ln} = \frac{\sqrt{(x_n - x_0)^2 + (y_n - y_0)^2}}{\sum_{i=0}^{n} \sqrt{(x_i - x_{i-1})^2 + (y_i - y_{i-1})^2}} \tag{5}$$

*Pen Lift Time* is the amount of time during each segment that the participant lifts their pen. Although participants are required to leave their pen on the tablet at all times as per the instructions of the Trail-Making Test, some participants still absent-mindedly lift the pen when searching for a dot or when correcting a mistake. This feature is intended to capture the behavior of both of these scenarios

to explore a possible correlation with MCI.

*Pen Pressure Average* and *Pen Pressure Standard Deviation* are features pertaining to the pressure that a participant places on the pen as they complete the test. We wanted to explore the possibility that a participant places more pressure on the tablet if they are unsure of their trajectory or if the test is difficult for them to complete.

We complete the feature set by adding a few sets from existing sketch and gesture recognition literature. We implemented 11 features from Long *et al.* [31] as a supplement to the Rubine features for general-purpose sketch recognition. Alamudun *et al.* [3] applied Rubine and Long features and added two direction-based features to help with saccade detection in an eye-tracking task, but have we believe can also be implemented as general-purpose sketch recognition as well. Finally, Paulson *et al.* introduced two features, normalized distance between direction extremes (NDDE) and direction change ratio (DCR), as general-purpose sketch recognition features that we also included for this study [42, 43].

### 5.4 Model Construction

Because Trail-Making Test behavior is characterized by the distinct actions of travelling to the next line and searching for the next, we decided to produce two separate classification models to explore the possibility of either being more indicative of MCI and compare their performance. Additionally, because the actions yield different behaviors, not all features were applicable for both types of actions. For example, line direction is important for travel lines to identify incorrect line deviation after we normalize travel lines as shown in Fig. 5. However, search lines cannot be normalized since direction at entry and at exit of a dot, even for healthy participants, depends heavily on the test layout itself. Table 3 lists every feature initially integrated into the feature set, and the subscripts next to the feature names indicate which were chosen for the models.

Some features were also removed from search and travel classification models due to a high collinearity value ($> 0.90$). Fig. 11 shows the collinearity heatmap of the remaining features that were used for both search and travel lines. Further, all values were normalized between 0 and 1.

Every segmented line from the 149 tests is included and is given the label according to the participant's cognitive state (MCI or healthy). 3,490 search lines and an equal number of travel lines were used for their respective classification models.

Models were constructed according to a 90/10 split for a 10-fold cross-validation. The models were trained and evaluated according to the two labels of MCI or healthy assigned during the screening phase of the study. We used 7 binary classification models to compare performance.

### 5.5 Prediction of MoCA Scores

Participant labels of "healthy, without MCI" and "healthy, with MCI" essentially is dividing participants between two broad categories of MoCA scores. Section 5.1 specifies the categories are a MoCA score of 26 and above for "healthy, without MCI" and between 19 and 26 for "healthy, with MCI". This in effect means our classification attempts to predict a wide range of the MoCA scores of the participants. However, we also sought to more directly predict the MoCA score in a more granular fashion as part of the data analysis of this study.

Our approach to MoCA score prediction is similar to the prediction of broad categories in that we are using the same training and classification features, and the same 90/10 split for 10-fold cross-validation. The similarity also extends to the training and classification being performed on the individual lines, and the F1-score and accuracy being calculated on how closely each **line** is being predicted to the actual MoCA score associated with that line's entire test. This is distinct from other methods of classification that seek to analyze the entire page and create a single prediction. We believe

Recursive Feature Elimination vs. Accuracy for MoCA Prediction (Search)

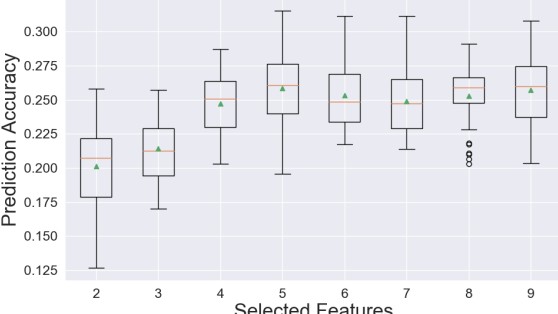

Figure 9: Box plot of amount of features chosen for Recursive Feature Elimination (search lines).

Recursive Feature Elimination vs. Accuracy for MoCA Prediction (Travel)

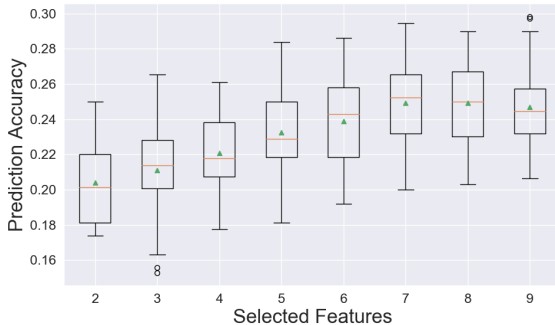

Figure 10: Box plot of amount of features chosen for Recursive Feature Elimination (travel lines).

that participants of Trail-Making Test do not perform evenly through the entirety of the test; while they might perform well for a few dots, a single dot might prove to be difficult for participants to find. Indeed, several participants from our empirical observations who performed poorly would find certain dots easy while finding others significantly more difficult. Our wish to capture this particular type of behavior is the reason behind our use of the segmented lines. We believe per-line sketch analysis and predictions might yield a novel insight into the participants' behavior. Because the original scoring system was conceived at a time when granular sketch analysis was not possible, we believe per-line analysis can provide a more granular and complete picture of a participant's behavior during the test. Prediction was trained and tested on both the Travel and the Search models.

We performed recursive feature elimination (RFE) on the Travel and Search Models to determine the top-ranking features that can be included in a logistic regression to predict the MoCA scores. The number of features ideal for both was determined to be 6 since that is the number of features where the accuracy plateaus for both Search and Travel models. The features selected for the Search and Travel models are listed in Table 4, and the box plot depicting the comparison of feature selection to accuracy is shown in Figures 9 and 10.

A logistic regressor with an iteration of limit of $n = 100000$ was employed for both models to predict the MoCA scores based on the features selected by the RFE. We then used a repeated K-Fold cross-validator, with a 90/10 split repeated 3 times for a total of 30 comparisons in the calculation of the predictors. To gauge the performance of the predictions being made, we calculated the average Mean Absolute Error (MAE) and the Root Mean Squared

Table 3: Classification features. **Model** describes whether the feature was used for the model for classification of travel (T) or search (S) lines. Some features were excluded due to high collinearity and/or were inappropriate for a specific model. *fittsSteering* is a scaled and averaged combination of the features *fitts* and *steering*.

| Name | Model | Name | Model | Name | Model | Name | Model |
|------|-------|------|-------|------|-------|------|-------|
| rubine1 | T | rubine10 | | avgPressure | T+S | openness | T+S |
| rubine2 | T | rubine11 | | stdevPressure | T+S | boundBoxArea | |
| rubine3 | T+S | rubine12 | T+S | avgSpeed | T+S | logArea | T+S |
| rubine4 | T+S | rubine13 | T+S | stdevSpeed | T+S | rotRatio | T+S |
| rubine5 | S | fitts | | aspect | | lengthLog | T+S |
| rubine6 | | steering | T+S | curviness | T+S | aspectLog | T+S |
| rubine7 | | lineRatio | T | relativeRot | T+S | fittsSteering | |
| rubine8 | S | hesitation | T | densityMetric1 | | ndde | T |
| rubine9 | T+S | penLiftTime | T+S | densityMetric2 | T+S | dcr | |

Table 4: Features chosen by Recursive Feature Elimination to directly predict MoCA scores.

| Search Model RFE Features | Travel Model RFE Features |
|---------------------------|---------------------------|
| aspectLog | *avgPressure* |
| *avgPressure* | *avgSpeed* |
| *avgSpeed* | *rubine12* |
| *logArea* | *rubine13* |
| *rubine11* | *stdDevPressure* |
| *stdDevPressure* | *steering* |

Table 5: Root Mean Squared Error (RSME) and Mean Absolute Error (MAE) of predicted points of MoCA scores.

| Error Metric | Travel Lines | | Search Lines | |
|--------------|--------------|----------|--------------|----------|
| | Average | Std. Dev | Average | Std. Dev |
| RSME | 3.325 | 0.132 | 3.315 | 0.134 |
| MAE | 2.415 | 0.110 | 2.406 | 0.105 |

Error (RMSE) of the predicted vs. the actual MoCa test scores for all of the lines. In this prediction algorithm all of the segmented lines of test and training participants have been labeled their respective MoCA scores. Although MoCA is not typically labeled on a per-line basis, our experiment is to determine whether such a prediction can be accurately made on per-line granularity. MAE and RSME were both used to help determine the mean error between the predictions of the logistic regressions. Both of the Travel and Search prediction algorithms had RSME and MAE as well as their standard deviation calculations and are shown in Table 5.

# 6  RESULTS

## 6.1  Accuracy Metrics of MCI Prediction

The main results of model performance are reported on Table 6, meant to report on how well a classification model trained and tested on travel lines and search lines independently is able to identify whether the author of those lines had MCI or was a healthy participant. A total of eight different classification models, listed in the **Classifier** column on the table, were trained with the features listed in section 5.3. Results are reported for both the search line model and the travel line model, and we report the models' accuracy, F1-score, precision, and recall. For both travel and search lines, Table 6 shows that the best performing models were created using a Random Forest classifier. Additionally, pressure-related features had among the highest feature importances when analyzing drop-column importances for the random-forest classifiers.

## 6.2  Accuracy Metrics of MoCA Score Prediction

Two sets of metrics can be reported for the MoCA score prediction: the results of Recursive Feature Elimination and how the number of features affects the prediction accuracy, and the results of the average MAE and RSME of the predictions made on the test data. Prediction of the MoCA, as opposed to the prediction of MCI, is non-binary and more of a continuous set of data in nature. For this exercise we allowed fractions of numbers to be predicted, since our chief method of comparison is the calculation of RSME and MAE. Small discrepancies in MoCA scoring due to the inclusion of non-whole

fractions would be minor, if that were the chief difference between predicted and actual scores. The accuracy metrics are reported in Table 5.

## 6.3  Discussion

### 6.3.1  Mild Cognitive Impairment

One of the primary challenges in detecting MCI is the inherently subtle nature of changes. Research such as that of Zhang *et al.* [64] outline difficulties in formalizing behaviors that correlate significantly with the manifestation of MCI in the Trail-Making Test. Depending on the severity of cognitive decline and multiple factors in how MCI affects each participant, they may not find the TMT specifically that challenging. For that reason, it is generally believed that the TMT, while proven sensitive to MCI in many cases, is not alone the only tool needed to reliably detect MCI.

The results from the accuracy metrics of the travel and search lines supports the notion that detecting subtle levels of MCI is inherently challenging if only analyzing one test. In several of our observed cases, participants who we classified as just under our MCI threshold based on their MoCA score completed the test in a similar manner as a typical healthy participant. In these cases a clinical neuropsychologist would continue testing their patient with several other kinds of exams or use the Trail-Making Test to primarily to identify other conditions of cognitive decline. This differs from other digital sketch recognition problems where the exhibited behaviors are not subtle by nature, or if the goal is to differentiate between discrete shapes. Models for those problems typically result in much higher accuracy and F1-scores (closer to $> 0.9$) since the labels are more cleanly delineated.

Overall, we believe the results present a meaningful contribution on the analysis of MCI through the TMT, largely due to the analysis and model construction on a per-line basis. Our implementation refined steps to segment the sketches by integrating speed thresholds to identify when the participant has found the next dot in the process. Whereas previous work in analyzing digitized TMT sketch data tends to average behaviors over an entire test, we sought to leverage the high-granularity nature of sketch data to provide analysis on individual lines. Our contribution also extends to the normalization

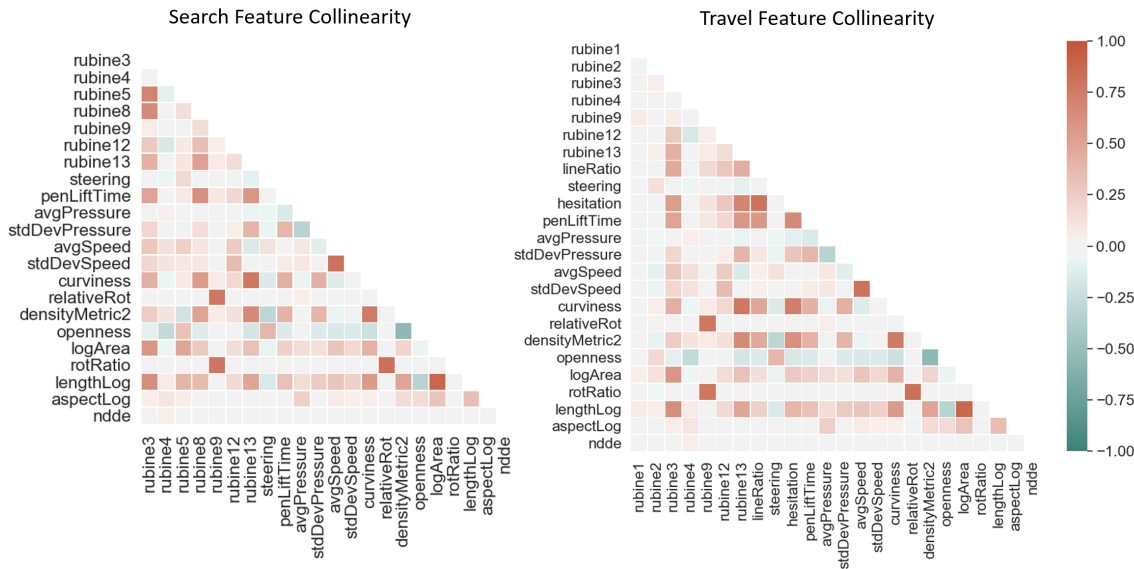

Figure 11: Feature collinearity for both search and travel lines. Features with collinearity above 0.9 were removed from the model

of line direction and total length to avoid differences between lines that are due to the TMT individual dot locations. The key is to eliminate potential confounders introduced by the fact that the TMT stimulates all participants to change line directions and total line length. We chose not to map a "perfect" line for each of the different segments to gauge performance, since Trail-Making Test layouts are numerous and clinicians frequently use modified versions for their own purposes. We sought to create a classification model that would work regardless of the dot layout to avoid creating a model that only works on that specific layout. Ultimately we sought to explore whether segmented lines could individually be labeled as MCI or healthy with at least similar performance as existing work.

A popular method for creating behavioral models is in the leveraging of deep learning techniques such as neural networks. These techniques are becoming more prevalent due to its ease of deployment for large datasets and higher efficacy in classification. However, we did not believe these techniques appropriate for this experiment for two primary reasons. The first is due to the necessity of collecting a considerably larger dataset for the creation of a classification algorithm using deep learning techniques. Challenges related to the proper collection of data for this experiment are explained in the following section. The second reason is due to the lack of explainability in deep learning techniques. While it would be possible to produce a more accurate behavioral model provided we acquire a considerably larger dataset, we would be unable to explain to a clinician which behaviors of the participant are responsible for the conclusion they are likely to have MCI. We believe that behavioral analysis in these types of domains should be usable to domain experts, thus motivating the manual creation of features to explain behavior.

We believe these results to be of interest in the HCI community, primarily due to the inherent nature of linking a cognitive examination with the analysis afforded by a high-granularity data collection protocol. In particular, the creation of an Index of Performance of sorts for the Steering Law (see Fig. 8) proved useful enough in both the search and travel line prediction models. For this particular project this calculation was different enough from Fitts' existing Index of Performance to warrant its inclusion as its own feature, and is potentially something that could be implemented to UX research. Indeed, we hope the results and explanations of the TMT can allow HCI researchers to see the TMT as a decades-old UI navigation task, and that the same principles and techniques that led to the creation of the Fitts' and Steering Laws can be applied to a digital TMT.

### 6.3.2 Montreal Cognitive Assessment Score Prediction

As previously mentioned, MAE and RSME outlined in Table 5 show our predictions for MoCA scores, both search and travel lines possessing a Mean Absolute Error of around 2.4 on average and a Root Mean Squared error of around 3.3. Essentially, regardless of whether the travel lines or search line data and features were used to predict the MoCA score of that individual user, the resulting error remained consistent. Although MoCA scores range from 0 to 30, our study ethics protocol prevented us from conducting research on participants with scores below 19 as previously mentioned, reducing the range of scores available to us to train and test on to between 19 and 30. The error rates reported implicitly become wider due to this range of scores being reduced, but we believe the reported MAE and RSME values still are small enough to be of interest to report. Overall, the scores suggest that the feature set presented in this paper can be used to predict MoCA scores based on a participants' digitized TMT sketch data.

Challenges of the MoCA score prediction were similar to those of predicting MCI, but were exacerbated by the labeling of a single score point to every line. Per-dot line segmentation resulted in likely an unbalanced training set, since a small subset of participants who performend fairly well the MoCA could skew the training and test sets considerably. This unevenness in MoCA distribution suggests that a much larger and wider range of MoCA scores is needed for accurate score prediction. As it stands, the Recursive Feature Elimination for both models as shown in Figure ?? suggests that even the optimal amount of chosen features yields only an accuracy above 0.35 for the Search model and an accuracy of up to 0.30. For this current version of the calculated features and those chosen by the RFE, we believe that additional features and changes to the existing ones would be necessary to increase the prediction accuracy.

At present the results for predicting MoCA scores are inconclusive. The errors as reported in Table 5 might suggest an average error of about 10% given the range of the MoCA scores to be from 0 to 30 points. The demographic data shown in Table 1 and discussed in section 5.1 shows an average MoCA score of 24.54 for all participants as well as the total overall criteria for inclusion of participants from 19 onward. Due to limitations on protocol safety, we are unable at present to recruit and test for participants with more severe cognitive

Table 6: Classification metrics. Acc is accuracy, F1 is F1-score, Prec is precision. For both the travel lines and search lines models, **n=3,490**

| | Travel Lines | | | | Search Lines | | | |
|---|---|---|---|---|---|---|---|---|
| Classifier | Acc | F1 | Prec | Recall | Acc | F1 | Prec | Recall |
| Majority | 0.51 | 0.51 | 0.50 | 0.50 | 0.53 | 0.53 | 0.52 | 0.52 |
| Gaussian Naive-Bayes | 0.47 | 0.36 | 0.60 | 0.53 | 0.47 | 0.38 | 0.58 | 0.53 |
| Decision Tree | 0.59 | 0.59 | 0.58 | 0.58 | 0.60 | 0.60 | 0.59 | 0.59 |
| K-Nearest Neighbor | 0.60 | 0.60 | 0.59 | 0.59 | 0.58 | 0.58 | 0.57 | 0.57 |
| Linear Regression | 0.65 | 0.64 | 0.65 | 0.63 | 0.62 | 0.59 | 0.61 | 0.58 |
| SVM | 0.65 | 0.63 | 0.66 | 0.62 | 0.63 | 0.61 | 0.64 | 0.60 |
| LDA | 0.65 | 0.63 | 0.65 | 0.62 | 0.62 | 0.60 | 0.61 | 0.59 |
| **Random Forest*** | **0.67** | **0.73** | **0.67** | **0.80** | **0.66** | **0.72** | **0.68** | **0.77** |

impairment for participants who score below 19. This is largely due to safety protocols requiring participants below that age to be accompanied by a guardian or healthcare official, since institutional review boards consider severely cognitive impaired individuals who would be unable to provide informed consent of their own volition. Following the safety protocols fortunately does not impair significantly the prediction of MCI vs. non-MCI populations since MCI participants are still able to provide informed consent, but this does reduce the efficacy of predicting MoCA performance as a continuous score. In order to create a more accurate MoCA predictor, we will require a larger corpus of data with a more even distribution of MoCA scores such that the scores are more evenly distributed as per established normative data. At present the inclusion of an MCI predictor did somewhat limit the performance of a MoCA score predictor.

## 7 LIMITATIONS AND FUTURE WORK

One of the main challenges in building an accurate predictive behavioral model is the creation of a new dataset for that specific purpose. Despite the fact that the Trail-Making Test has been in use for several decades, the granularity of digital data and the requirement of a digital pen necessitated the creation of a new dataset. The prevalence of different Trails Test layouts and the small differences of protocol that vary from clinician to clinician also necessitated a unified testing protocol. Accompanying this challenge is the laborious recruitment process. Although the task is simple, the administration of the MoCA and the proper administration of the Trail-Making Test resulted in a slower rate of data collection that is typical of sketch recognition tasks.

Currently, the age ranges of the Trail-Making Test's normative data, as found Tombaugh's stratified normative data for paper-and-pencil Trail Making Tests [61], divides the age range into 11 distinct categories. Our normative data covers the latter 7 bins with our participants ranging from 57 to 86 years of age. Since the focus of this experiment is in identifying MCI among middle aged and older individuals, the study focused on that age range. Future studies will continue the data collection process to build a more complete normative body of data across all age ranges. These might potentially result in differing behaviors between patients with MCI from different age ranges, but a solid body of data from those age ranges is necessary for verification. We also aim to further expand on localizing areas that were difficult for participants with MCI, reporting these lines on a UI-level in real time and evaluating a clinician's diagnosis experience with such an automated tool.

Although the system has two primary end users, the scope of this paper focused on the participant. We aim to investigate the user experience for proctors to deploy the system and use the predictions in their diagnosis. Specifically we aim to gather feedback on the experience of reporting the system's findings, since the proctors have access to a wide variety of sketch visualizations as mentioned in

Section 3. Reporting on predicting MCI and non-MCI participants in addition to highlighting hesitation, line deviation, and visually color-coding search and travel lines offers proctors a large range of information and future work will investigate on the usefulness and overall user experience. Additionally we would like to use other peripherals for additional features such as heart rate sensors and integrated eye-tracking solutions to create an even more feature-rich data set that enhances participant behavior analysis.

Also of note is the fact that a protocol of collecting data on an MCI population inherently removes a full range of ages and conditions for normative data. This impacts the ability for a predictive system to make an ML-based prediction of the actual MoCA score. The prediction of the MoCA scores yielded relatively small error percentages but when taking into account the reduced range of MoCA scores that were available for testing and training, we conclude the results for direct prediction of exact MoCA scores are inconclusive despite being somewhat promising. We are considerably more confident about the binary classification between non-MCI and MCI populations precisely because the range of data and the collection protocol yielded the most appropriate data for that kind of classification. Future work will require a wider range of participants with normative data closer to that of Tombaugh *et al.* [61]. We are confident the digital sketch data from digital TMTs can be used to make much more accurate predictions about the participants' MoCA scores if said data were available.

Subtle changes in behavior due to Mild Cognitive Impairment continue to present significant challenges in identifying the earliest possible signs for conditions that may lead to dementia and Alzheimer's disease. Existing efforts to aid in this challenge highlight the difficulty of finding the nuances in behavioral changes present in a Trail-Making Test. However, with significant improvements over previous efforts we present a solution that suggests individual lines, regardless of their direction, can distinguish between MCI and Healthy with noticeably higher levels of accuracy. We look forward to employing additional preprocessing methods, features, and a larger digital sketch dataset to further improve on this effort. We believe sketch data from the Trail-Making Test still has the potential to yield insights into behavioral changes that are yet to be discovered.

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
