# OpenReview forum: "Detecting Mild Cognitive Impairment Through Digitized Trail-Making Test Interface"
_graphicsinterface.org/Graphics_Interface/2022/Conference — GI 2022_

### Official Review · Reviewer_orF7 · 2021-12-23
**Interesting paper, however, it is encouraged to improve some parts.**

**Rating:** 6
**Confidence:** 4

**Review:**

The authors improved an existing interface of digitized TMT application which was aimed to provide the closest user experience to the traditional MCI diagnosis method. The paper also discusses how the proposed method can classify MCI from the drawing data as well as the accuracy of prediction.
In general, the paper provided a well-structured literature review to provide the introductory background to understand their research. Even though the proposed method uses a different score system to detect MCI, the authors stated that their interface was built on top of existing work. The authors clearly stated their contribution and were well investigated throughout the paper.

Comments
- MOCA score range could be described for a better understanding of the background. The range interpretation was mentioned, however, the maximum or minimum possible scores were not described.
- In Table 1 caption, the CI reporting should be corrected for a better description. Having an age of 0.94 sounds ambiguous and incorrect.
- In equation 1, why was the constant c = 40?
- Why was hesitation determined for a speed less than 0.4? Why not 0.3? Or 0.5?
- One future work would be to allow recording of process in drawing/TMT. One advantage of digitization would be to ‘record’ the path, and recording the animated process and exporting the final video/file would be useful for later analysis. Another advantage of using digitized devices would be using eye trackers, which could generate useful data such as hand-eye coordination.
- It would be important to address how possible bias has been controlled. For example, data collection on a specific group would cause data bias, which may be discussed/controlled when creating a prediction model.
- Authors claimed to have low accuracy in prediction, and that could still be promising for future open possibilities. It could be interesting if the actual users of the system (i.e., practitioners) could evaluate the system rather than limit the data analysis in empirical methods.

---

### Official Review · Reviewer_HfVi · 2022-01-14
**important work, but more groudning in the HCI literature needed**

**Rating:** 6
**Confidence:** 4

**Review:**

This paper presents a digital pen-based version of the Trail-Making Test (TMT), which is a test that is used (in conjunction with other tests) to detect mild cognitive impairment (MCI). The key contribution is the way in which the gestures are segmented (based on individual lines) and the ML models that are created which show (what I understand to be) good precision and recall.

There is a lot that I like about this paper. Turning clinician-administered and scored cognitive tests into ones that can be semi- or fully-automatically scored is an important area of research. Many of these tests are relatively simple on the surface, but the assessment can be nuanced. Although not mentioned in the paper, it is my understanding that a test such as the MoCA, can be administered and scored by someone with relatively little training. The tests that make up neuropsychological battery (including the TMT), require a trained clinician. In some places in the world, there can be significant wait times to getting tested. Given the aging population, it is imperative that we lower the barriers to testing. Automating the scoring of the test should help.

As best I can tell, the model work was well done. ML modelling is not my area of expertise though.

My challenge in reviewing this paper is that it does not feel as though it is written for an HCI audience, and more specifically to make an HCI contribution. While it certainly leverages HCI research, such as the Steering Law and gesture recognition, the Introduction, Related Work, and the Discussion do not seem to be framed for an HCI audience. Typically an Introduction will end with the claimed contributions of the work. In this paper they don’t come until Section 2.3, and work presented in 2.2 is critical for framing the “per line” contribution. The Results are quite short for an HCI paper. I personally would have benefited from some unpacking of Table 6.  The Discussion is quite narrowly about the model, but I am left wondering what HCI researchers can take away from this work in addition to the model?

In particular, here are the things I wondered about:
-	It is known that older adults can struggle with pressure, both a stylus and direct touch. Shouldn’t prior experience with digital technologies have been taken into account in the research design? Given the small sample size, what if a larger proportion of participants with MCI had lower technology experience? This is in relation to the reporting that pressure-related features were most important in the model.
-	There has been some HCI research in cognitive testing and interface design. While it does not negate the contributions being claimed in the current paper, it would certainly help to ground the importance of research reported in the paper for the HCI audience.


Sean-Ryan Smith. 2017. Mobile context-aware cognitive testing system. In Proceedings of the 19th International Conference on Human-Computer Interaction with Mobile Devices and Services (MobileHCI '17). Association for Computing Machinery, New York, NY, USA, Article 73, 1–4. DOI:https://doi.org/10.1145/3098279.3119926

Sean-Ryan Smith. 2016. Dynamic Online Computerized Neuropsychological Testing System. In Companion Publication of the 21st International Conference on Intelligent User Interfaces (IUI '16 Companion). Association for Computing Machinery, New York, NY, USA, 118–121. DOI:https://doi.org/10.1145/2876456.2876460

Matthew Brehmer, Joanna McGrenere, Charlotte Tang, and Claudia Jacova. 2012. Investigating interruptions in the context of computerised cognitive testing for older adults. In Proceedings of the SIGCHI Conference on Human Factors in Computing Systems (CHI '12). Association for Computing Machinery, New York, NY, USA, 2649–2658. DOI:https://doi.org/10.1145/2207676.2208656


-	Are there other areas of HCI work that would benefit (or have benefited) from a per-line segmentation of a task or the adaptation to the Steering Law that was used in this work?
-	In the line direction normalization, was there any consideration to the fact that movement in all directions is not equally difficult for users?
-	At the end I felt like a broader reflection on how this approach might eventually be used with a “clinician in the loop” or for full self-administration would benefit readers.

In looking up the citations above from the ACM DL when finishing up this review, I noted this recent paper from GI specifically on the TMT, which is not cited and seems quite relevant/important:

Raniero Lara-Garduno, Takeo Igarashi, and Tracy Hammond. 2019. 3D-Trail-Making Test: A Touch-Tablet Cognitive Test to Support Intelligent Behavioral Recognition. In Proceedings of the 45th Graphics Interface Conference on Proceedings of Graphics Interface 2019 (GI'19). Canadian Human-Computer Communications Society, Waterloo, CAN, Article 10, 1–8. DOI:https://doi.org/10.20380/GI2019.10

It is instructive that the framing in the Introduction and  Related Work for that paper and the framing more generally is more consistent with an HCI paper.

This one also popped-up, seems important and is not cited:

Alexander Prange and Daniel Sonntag. 2021. Assessing Cognitive Test Performance Using Automatic Digital Pen Features Analysis. Proceedings of the 29th ACM Conference on User Modeling, Adaptation and Personalization. Association for Computing Machinery, New York, NY, USA, 33–43. DOI:https://doi.org/10.1145/3450613.3456812

Without grounding the contribution in the HCI more fully than just gesture recognition, it is difficult for me to argue more strongly to accept  this paper.


Minor:
-	There are quite a few typo kinds of issues throughout (like repeated or missing words), suggesting that the paper was not proofread
-	Please clarify which Center for Disease Control and Prevention is referred to in order to ground the figure of 5 million Alzheimer’s patients (I assume it is the US)
-	“A total of seven different classification models, list in the Classifier column on the table…” I see eight listed in that table.
-	What is the difference between fitts and steering AND fittsSteering in Table 3?
-	Please clarify the duration of the test. It seems they started at 8AM and they did a short test in the morning following and then another one in the afternoon. That seems like a very long test day, especially for older adults.
-	“hover” is often used to refer to the “hover space” which is an area above the surface where a stylus can still be detected. This could easily be confused with the use of “hover” in the paper.
-	I found the clarity around “user”, “proctor”, “participant” somewhat confusing at first.
-	Index of Performance is mentioned, but it is Index of Difficulty that is given (equation 2)


I hope these comments are helpful and wish the authors success with their work.

---

### Official Review · Reviewer_uDQz · 2022-01-17
**A Useful Exploration of Techniques for Detecting MCI on TMT**

**Rating:** 7
**Confidence:** 4

**Review:**

This paper is, essentially, an exploratory study where the authors ask whether a model of performance on a digital trail-making-task can be built that allows reliable detection of mild cognitive impairment. The answer is "perhaps, with different features of a different model", but even with the ambiguous results, I feel the paper is a useful contribution to the state-of-the-art.

This paper provides a useful structured overview of the goals, motivations, and current state-of-the-art. The techniques uses and the rationales are clearly explained. The approach, while not extremely innovative, represents a well-executed evaluation that is novel to the best of my knowledge, and highlights both the potential and current shortcomings of the model developed.

I found the model interesting. It makes use of 13 features from Rubine's features for geometric unistroke shape recognition. While this work is over 30 years old, the general purpose nature of these features as a stroke descriptor was a really interesting attempt at classification on the TMT. Obviously, a significant part of the discussion focuses on the failures of the model and the rationale for those failures, but, even with the limitations of negative results, I find this work is worthwhile. It highlights a premise (Can we use simple features of a route drawing task to correlate with the Montreal Cognitive Assessment or to predict Mild Cognitive Impairment?), it analyzes the results (F-scores for classification and RMSE for correlation/prediction -- note that it is RMSE and not RSME as in Table 6). -- which are the appropriate metrics), and it presents avenues for future work (better models, e.g. NN with greater data sets or better features, e.g. beyond Rubine for the current models explored).

In the end, this paper represents, in my opinion, a very useful paper for a graduate student interested in these issues to read.

---

### Decision · Program_Chairs · 2022-01-18

Accept